# The Effect of MQL on Tool Wear Progression in Low-Frequency Vibration-Assisted Drilling of CFRP/Ti6Al4V Stack Material

Ramy Hussein [1],*, Ahmad Sadek [2], Mohamed A. Elbestawi [3] and Helmi Attia [2]

[1] Department of Mechanical Design and Production, Technical Research Center, Cairo 11461, Egypt
[2] Aerospace Manufacturing Technology Centre, National Research Council Canada, Montreal, QC H3T 2B2, Canada; Ahmad.Sadek@nrc-cnrc.gc.ca (A.S.); helmi.attia@mcgill.ca (H.A.)
[3] Department of Mechanical Engineering, McMaster University, Hamilton, ON L8S 4L7, Canada; Elbestaw@mcmaster.ca
* Correspondence: Husser2@mcmaster.ca or eng.roma27@gmail.com; Tel.: +20-111-701-8880

**Abstract:** In this paper, the tool wear mechanisms for low-frequency vibration-assisted drilling (LF-VAD) of carbon fiber-reinforced polymer (CFRP)/Ti6Al4V stacks are investigated at various machining parameters. Based on the kinematics analysis, the effect of vibration amplitude on the chip formation, uncut chip thickness, chip radian, and axial velocity are examined. Subsequently, the effect of LF-VAD on the cutting temperature, tool wear, delamination, and geometrical accuracy was evaluated for different vibration amplitudes. The LF-VAD with the utilization of minimum quantity lubricant (MQL) resulted in a successful drilling process of 50 holes, with a 63% reduction in the cutting temperature. For the rake face, LF-VAD reduced the adhered height of Ti6Al4V by 80% at the low cutting speed and reduced the crater depth by 33% at the high cutting speed. On the other hand, LF-VAD reduced the flank wear land by 53%. Furthermore, LF-VAD showed a significant enhancement on the CFRP delamination, geometrical accuracy, and burr formation.

**Keywords:** vibration-assisted drilling; low-frequency vibration-assisted drill; CFRP/Ti6Al4V; stacked material; minimum quantity lubricant; tool wear; surface integrity; delamination; burr formation

## 1. Introduction

The rapid advancements in the new generation of aircraft explains the growing usage of lightweight materials in the hybrid structure design [1]. This structure commonly consists of carbon fiber-reinforced polymer (CFRP) and the Ti6Al4V titanium alloy in different coupling forms such as CFRP/Ti6Al4V, CFRP/Ti6Al4V/CFRP, and Ti6Al4V/CFRP/Ti6Al4V [2,3]. Compared to the usage of the uni-material structure, the hybrid design showed a superior physical and mechanical properties such as a high strength-to-weight ratio, low coefficient of thermal expansion, corrosion/erosion, and fatigue resistance [4,5]. The mechanical fastening using rivets and bolts is commonly used during the assembly process of these materials. Consequently, the single drilling process of the CFRP/Ti6Al4V stack materials has been identified as an effective method to achieve high precision assembly and productivity.

A swift tool wear progress was identified as one of the major challenges during the practical one-shot drilling process of CFRP/Ti6Al4V stack materials [6,7]. This tool wear progression can be described by two main mechanisms: abrasive wear of the hard carbon fiber particles [8], and an adhesion wear mechanism during the drilling process of Ti6Al4V [9]. The machining process of Ti6Al4V resulted in a significant increase in the cutting tool temperature due to the low thermal conductivity, and the continuous chip morphology of Ti6Al4V [10,11]. The high cutting temperature has a severe effect on the cutting tool condition, particularly during the drilling process where the poor evacuation mechanism located. Consequently, this process has a high tendency for chip accumulation and the tool–chip welding takes place [9,10]. Furthermore, by increasing the cutting tool

temperature, the tool material suffers from low hardness properties. Hence, the one-shot drilling process of CFRP/Ti6Al4V will show a severe abrasion wear mechanism from the CF particles. Thus, reducing the cutting temperature and segmenting the titanium chips have been emphasized by several studies.

Conventional drilling (CD) parameter optimization [12], pick-up drilling [13], orbital drilling [14,15], and vibration-assisted drilling (VAD) [16,17], are some of the main strategies that have been investigated to reduce the cutting temperature and change the titanium chips' morphology. Based on machining productivity, the VAD is the most promising machining method for stacked materials. In VAD, an axial tool oscillation is superimposed on to the normal tool feed direction, to generate an interrupted drilling process [18]. The interrupted cutting resulted in substituting the continuous Ti6Al4V chip morphology by segmented chips with a controllable geometry [9]. This achievement resulted in a better chips evacuation efficiency with a preferable cyclic cutting/cooling duty [9]. Thus, lower cutting temperature, better geometric accuracy, high surface quality, and compressive residual stresses can be induced [9].

Despite the promising results of LF-VAD in terms of tool durability, the presented investigations failed to discuss two main factors: the machining productivity and the associated effect of tool wear progression on the hole quality. The investigated range of cutting speed was limited by 20 m/min [19,20], without any clarification of the associated hole quality that represents the critical criteria for the aerospace industry. In a previous study [10], the effect of LF-VAD on tool wear and the associated hole quality was presented using cutting speeds up to 56.52 m/min, which is three-fold the illustrated range, using a recommended vibration amplitude [21]. The study was conducted using a dry coolant condition that resulted in reduce the cutting tool temperature by up to 40%. However, the dry coolant condition showed an unsuccessful drilling process of 50 holes that attributed to the observation of the tool–chip welding process.

The utilization of LF-VAD with forced air resulted in decrease the cutting temperature by up to 51.3%, enhance the chips evacuation efficiency, and reduces the tool–chip welding probability [19]. Despite the slight thermal load reduction compared to the dry coolant, the presented study was limited by a 20 m/min cutting speed which had a negative impact on the machining productivity. Furthermore, the author did not show the effect of forced air medium on the machined surface quality. Hence, the usage of a proper coolant medium could result in a substantial enhancement on the cutting tool life and machining performance.

Cryogenic cooling is a technique used to generate a high temperature gradient between the cutting zone and the coolant medium. This thermal gradient results in a higher machining performance through a proper cooling mechanism. Compared to the dry coolant, the assessment of $CO_2$-cryogenic coolant during the drilling process of CFRP/Ti6Al4V stacked, the geometrical analysis showed a great enhancement for 160 hole drilling process [22]. The utilization of $CO_2$-cryogenic coolant reduced the cutting temperature to lower than 26 °C at the Ti6Al4V exit surface. This reduction resists the tool thermal expansion and eliminates the adhesion process of titanium chips. Despite using low magnitude machining parameters, the presented study showed a significant increase in the cutting power consumption, and the author did not show any tool wear analysis. Recently, more studies have focused on the effect of cryogenic cooling during the machining process of CFRP and Ti6Al4V separately [23]. A novel method of a cryogenic cooling strategy (between-the hole) has been investigated [24]. Based on five drilled holes, the cryogenic cooling resulted in a lower flank wear with enhanced machining surface roughness, while the cutting forces and the specific cutting energy did not show a significant change. Moreover, the study did not show an economical solution in terms of production time and cost. Cryogenic cooling showed a promising result in terms of machining performance of difficult to cut material and eliminating the tool wear progression. However, most of the cryogenic systems in the literature cannot be classified as a commercial system. These systems require some machine adaptation, which cannot be used for the high productivity industry [23]. Furthermore, the

high temperature change of the cutting tool and machined materials will suffer from the low ductility and fragility [22]. These phenomena are crucial during the drilling process of VAD, where a repetitive impact mechanism at the tool chisel edge is predictable.

Minimum quantity lubricant (MQL) is a technology used to approach the tool/workpiece interface with a minimum amount of coolant medium to achieve the economic and environmental profits [25]. In terms of cutting temperature, tool life, and machining performance, the utilization of MQL showed a significant enhancement during the CD and LF-VAD process of CFRP/Ti6Al4V, as presented in [25,26]. The CD of CFRP/Ti6Al4V using MQL coolant, showed a great reduction in the frictional force at the tool–hole wall surface [27]. This reduction significantly reduced the specific cutting energy. Lower cutting energy combined with cutting temperature resulted in a proper machining performance in terms of geometrical accuracy and burr height. However, the machining process still suffers from the continuous titanium chips evacuation mechanism.

On the other hand, the combined effect of LF-VAD and MQL coolant showed a 42% reduction in the cutting temperature with a long tool life. This achievement could be the first step toward a high process stability for the automated drilling operation. However, a comprehensive study using a high cutting speed range and presenting the associated effect on the machining performance is highly recommended.

The current study presents the effect of LF-VAD under MQL coolant medium on the tool wear progression and the associated hole quality. The tool analysis covers the tool flank surface, rake surface, and the chisel edge. This analysis was correlated to the kinematics of VAD for a better understanding. Furthermore, the study presents a clear comparison between the effect of LF-VAD in dry and MQL conditions, for the same machining parameters.

## 2. Experimental Setup

All the experiments were carried out on a five-axis CNC machining center (Makino A 88), with the MITIS tool holder PG8045B3_HSK-A100_ER40 [28]. This tool holder has a fixed 2.5 oscillation/rev modulation frequency with a variable amplitude selection range (0.01 to 0.48 mm). The cutting tool temperature was monitored at the exit surface using the FLIR SC8000 HD series infrared camera (FLIR® Systems, Inc., Burlington, Canada), with a special support, as shown in Figure 1. As recommended for the aerospace manufacturer, A YG-1 Canada group [29], a 6 mm uncoated tungsten carbide (WC) twist drill was utilized during the conducted test matrix. The tool had a 118°-point angle with coolant through for MQL supply. The workpiece material had a CFRP/Ti6Al4V stack sequence with a 120 mm side length. The CFRP had a total thickness of 5.8 mm, while the Ti6Al4V plate was produced according to the grade 5 standard with a 6.75 mm thickness. Based on the previous experimental investigation [10,21]. Table 1 presents the selected machining parameters, tool, and workpiece materials. The test matrix consisted of two levels of cutting speeds (2000 and 3000 rpm), three levels of vibration amplitudes (0.1, 0.16, and 0.25 mm), and a fixed feed rate of 0.075 mm/rev. The selected test matrix parameters were targeted to overcome the previously published machining issues, as presented in [10]. The applied minimum quantity lubricant (MQL) has a 5% concentration of the MECAGREEN 550 with a 100-bar pressure, as recommended by the Canadian National Research Council (CNRC), Aerospace Manufacturing Group.

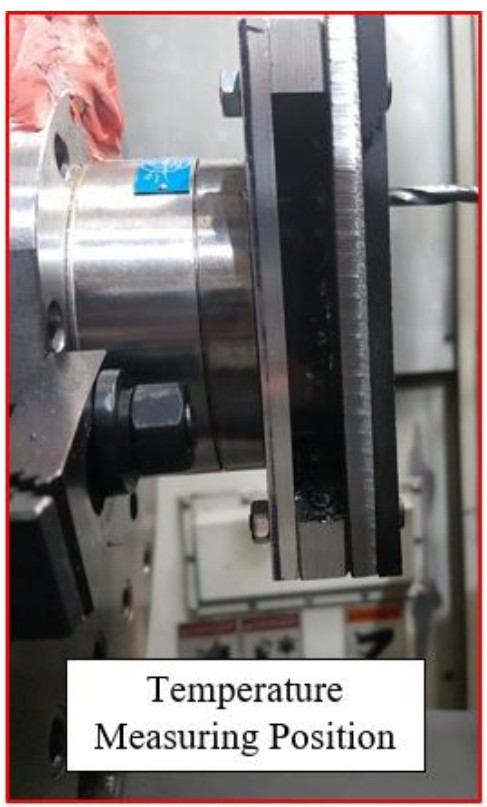
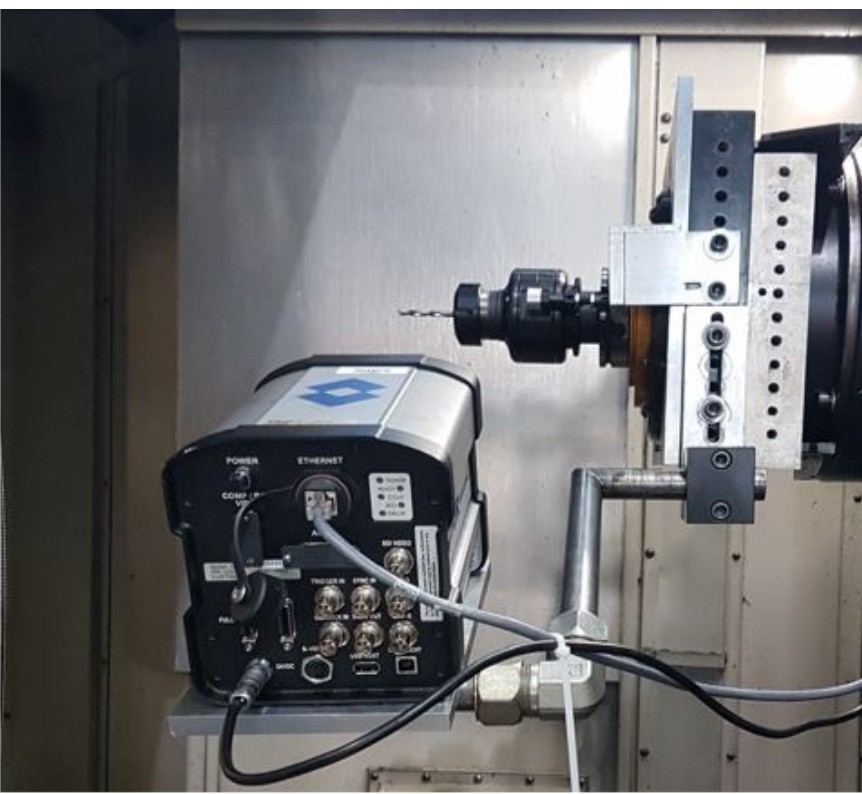

**Figure 1.** The experimental setup and workpiece materials.

**Table 1.** Specification of machining parameters, tool, and workpiece.

| Machining Parameters | |
|---|---|
| Cutting speed, N | 2000 and 3000 rpm |
| Feed rate, f | 0.075 mm/rev |
| Amplitude, Am | 0.1, 0.16 and 0.25 mm MQL |
| Cooling medium | |
| **Cutting Tool** | |
| Material | Tungsten Carbide |
| Diameter | 6 mm |
| Point angle | 118° |
| Helix angle | 20° |
| Manufacturer | YG-1 |
| **Workpiece material specification** CFRP Titanium alloy Stacking sequence | - 5.8 $\pm$ 0.02 mm of 42 $\times$ L-930(GT700) woven plies with the configuration $[[0,90]_{21}]$s, and flame retardant modified epoxy prepreg. <br> - Decomposition temperature is 320 °C. <br><br> Ti6Al4V grade 5 <br> CFRP 5.8 $\pm$ 0.02 mm/Ti6Al4V 6.75 $\pm$ 0.02 mm |

The tool wear analysis was based on a continuous drilling process of 50 drilled holes. The Winslow engineering tool analyzer model 560 was used for the rake and flank surface examination (Winslow engineering Inc., Peebles Lane, WI, USA). To investigate and identify the tool wear mechanisms, TESCAN VP.SEM scanning electron microscopy (SEM) and energy-dispersive X-ray spectroscopy (EDS) were used (Vega 3-TESCAN, Brno, Czech Republic). The rake face surface topography was inspected using the Alicona instrument (Alicona Manufacturing Inc., Bartlett, IL, USA). The CFRP delamination was analyzed

using the Keyence digital microscope VHX-6000 series (Keyence Corp., Osaka, Japan). The Mitutoyo coordinate measuring machine (CMM) model MACH 806 (Mitutoyo company, Kawasaki, Japan), was used to measure the drilled hole diameter error based on the average of ten points on two surfaces of each material.

## 3. Kinematics of VAD

Discarding the non-cutting machining process under the chisel edge through the axial tool oscillation for the vibration-assisted drilling, it showed a recommended enhancement during the drilling process of CFRP/Ti6Al4V, as presented in [21]. Based on the instantaneous cutting edge trajectory as described in Equations (1)–(3) [30], the effect of the vibration amplitude and cutting speed on the axial tool velocity has been investigated.

$$Z_1 = \frac{\gamma}{360°} * f + A_m * \sin(\gamma * F) \tag{1}$$

$$Z_2 = \frac{(\gamma - 180°)}{360°} * f + A_m * \sin[(\gamma - 180)*F] \tag{2}$$

The dynamic uncut chip thickness could be expressed as:

$$t_0 = \begin{cases} Z(\gamma) - \max(Z_k(\gamma)) & , \quad Z(\gamma) > \max(Z_k(\gamma)) \\ 0 & , \quad otherwise \end{cases} \tag{3}$$

where max $(Zk(\gamma))$ represents the maximum height of the preceding rotation, $(\gamma)$ represents the rotational angle, and to represents the dynamic uncut chip thickness.

During the drilling process of LF-VAD, increasing the vibration amplitude showed an obvious change on the chip formation. As shown in Figure 2, increasing the vibration amplitude resulted in increasing the dynamic uncut chip thickness, while the chip radian reduced. Based on Equation (3), the calculated uncut chip thickness was 0.15 mm for all vibration amplitudes, which is double the programmed feed. However, increasing the $A_m$ resulted in reducing the chip radian by 30%. The chip radian reduced from 65° at $A_m = 0.1$ mm to 45° at $A_m = 0.25$ mm. This reduction had a direct impact on the chip evacuation mechanism and consequently, the cutting tool temperature and the tool wear mechanism should be enhanced during the drilling process of the Ti6Al4V layer.

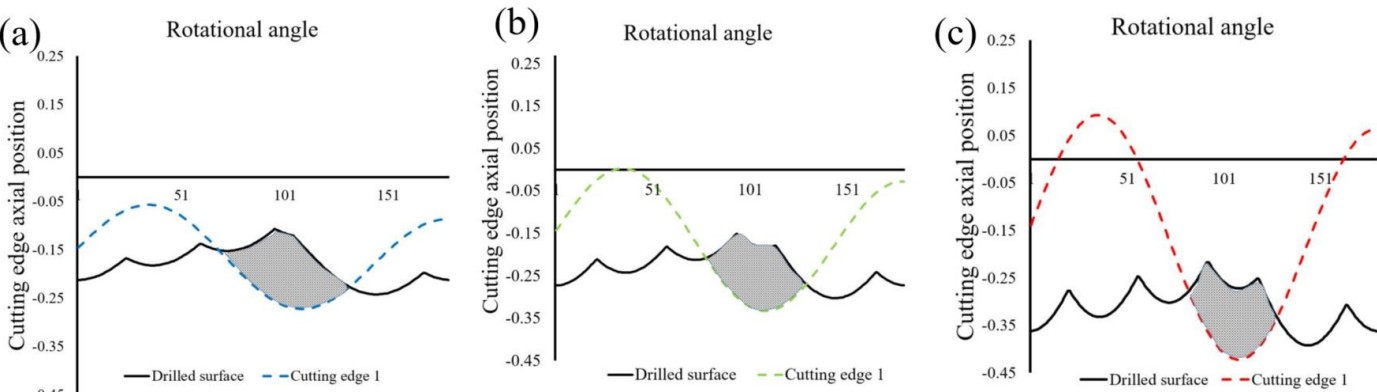

**Figure 2.** Effect of vibration amplitude on the uncut chip formation at (**a**) $A_m = 0.1$ mm, (**b**) $A_m = 0.16$ mm, and (**c**) $A_m = 0.25$ mm.

Figure 3 presents the axial tool velocity at different vibration amplitudes and cutting speeds. The vibrational amplitude was identified as the dominant effective parameter controlling the axial tool velocity followed by the cutting speed. Increasing the axial tool velocity would result in a proper cooling environment, low exit delamination factor, and higher evacuation efficiency. However, this increasing could result in severe chisel edge damage due to the repetitive tool–workpiece impact mechanism. Hence, chisel edge

inspection is highly recommended to identify the vibration amplitude threshold at the onset of chisel fracture.

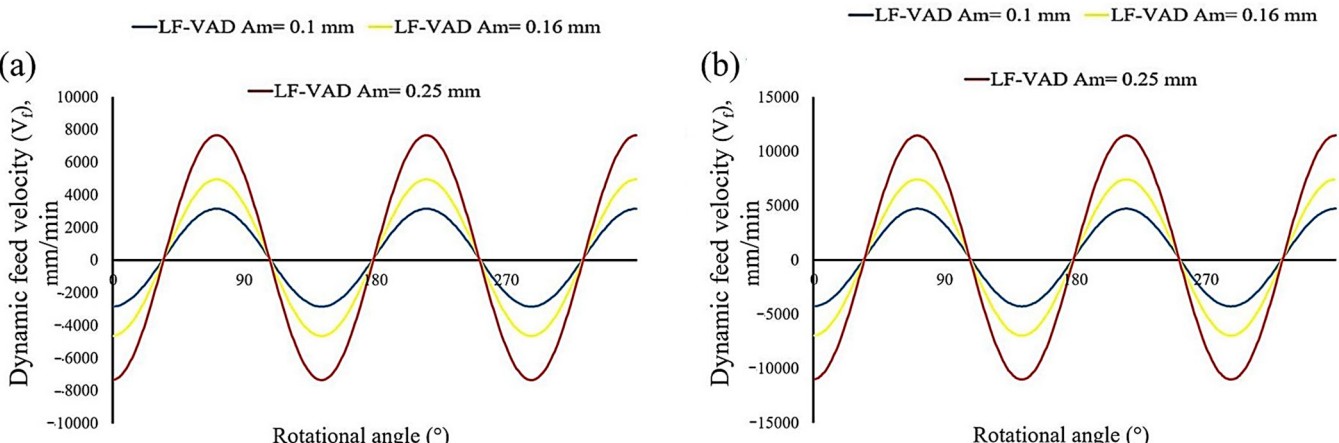

**Figure 3.** Effect of vibration amplitude ($A_m$) on the axial tool velocity at different cutting speeds (**a**) $N$ = 2000 rpm (**b**) $N$ = 3000 rpm.

## 4. Results and Discussion

### 4.1. Cutting Temperature

The effect of tool wear progression on the cutting tool temperature was measured at the exit surface of the Ti6Al4V layer for different cutting speeds and vibration amplitudes, as shown in Figure 4. For all machining conditions, LF-VAD showed a significant reduction in the cutting temperature compared to the CD. This reduction percentage was increased from 55% for $A_m$ = 0.1 mm to 65% for $A_m$ = 0.25 mm at $N$ = 2000 rpm, and from 45% for $A_m$ = 0.1 mm to 68% for $A_m$ = 0.25 mm at $N$ = 3000 rpm. This reduction can be described through two mechanisms: (i) the kinematics effect of VAD, (ii) the advance of MQL coolant. Firstly, as described in Section 3, increasing the vibration amplitude resulted in reducing the chip radian by 30% and increasing the axial tool velocity. The chip radian reduction enhanced the chip evacuation mechanism, while increasing the axial tool velocity significantly reduced the tool–chip contact time. Consequently, increasing the vibration amplitude has a positive impact in the cutting tool temperature, which is in agreement with the results presented in [9]. Moreover, high vibration amplitude resulted in reducing the duty cycle and increased the cooling cycle [9]. Secondly, the utilization of MQL led to the creation of a proper coolant medium, and facilitated the chip evacuation mechanism through the pressurized coolant, as presented in [31,32].

Compared to the CD at $N$ = 2000 rpm, the analysis of the results showed that the cutting tool temperature under LF-VAD was reduced by 40% for the dry coolant condition [10], and 65% for the MQL coolant condition, respectively, as shown in Figure 5. For high cutting speed ($N$ = 3000 rpm), tool–chip welding was confirmed as the main issue that resulted in stopping the drilling process at the second hole for CD, and hole number 20 for LF-VAD with $A_m$ = 0.25 mm, as reported in [10]. The usage of MQL resulted in a successful machining process of 50 drilled holes without any observation of the tool–chip welding process. This achievement was traced back to the significant reduction in the cutting tool temperature, lower tool–chip friction, and the higher chip evacuation efficiency as a secondary effect of MQL. Moreover, the utilization of MQL showed a clear reduction in the frictional force at the tool–hole wall interface, which reduced the specific cutting energy, as presented in [27]. Compared to the dry CD, the usage of MQL created smaller micro droplets to penetrate the tool–chip and tool–workpiece interface, coolant that resulted in a proper cooling mechanism [32].

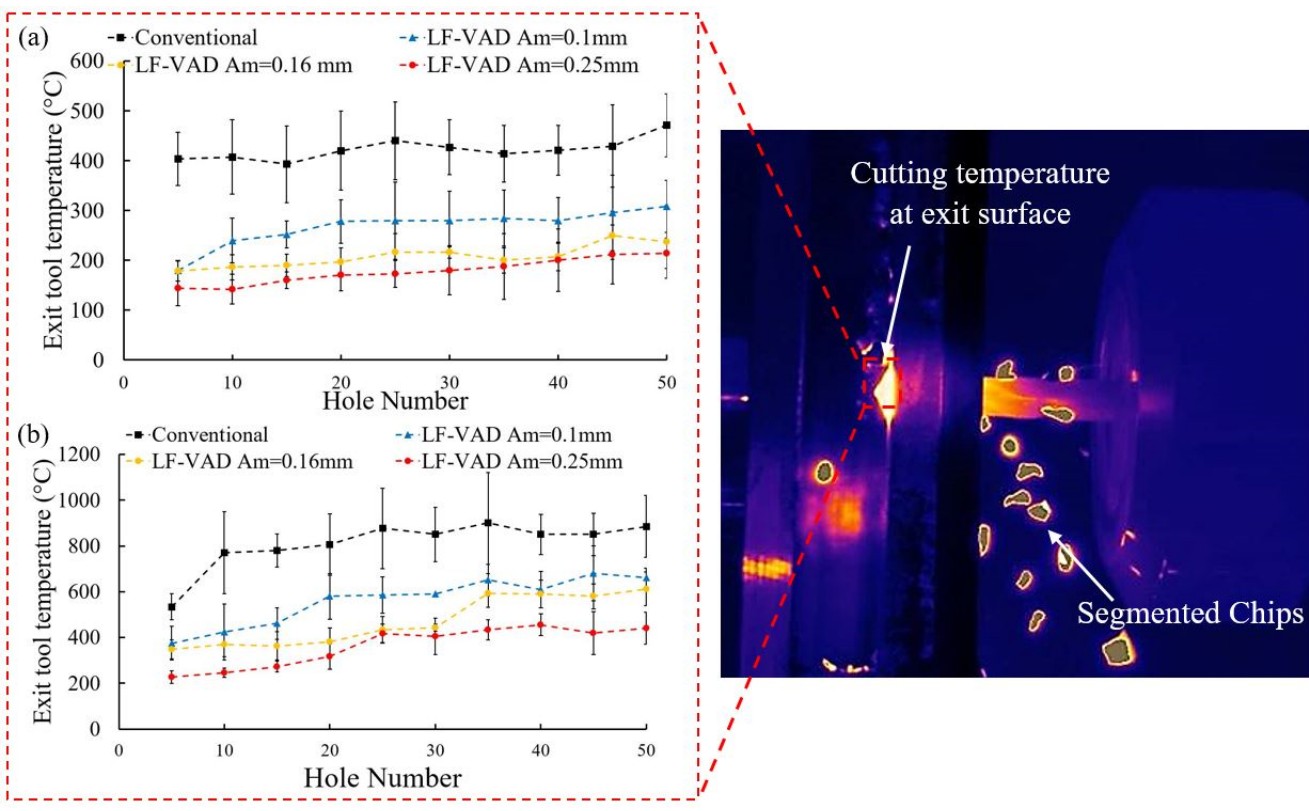

**Figure 4.** Effect of the tool wear progress on the tool cutting temperature at the exit surface for (**a**) *N* = 2000 rpm, and (**b**) *N* = 3000 rpm.

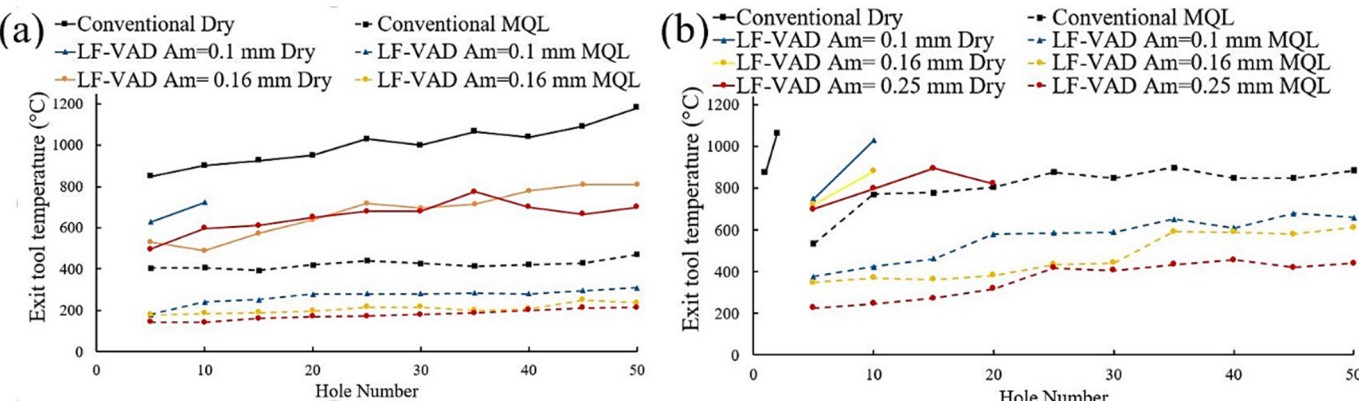

**Figure 5.** Effect of coolant medium on the cutting temperature for (**a**) *N* = 2000 rpm (**b**) *N* = 3000 rpm.

### 4.2. Tool Wear Progression and Mechanism

#### 4.2.1. Flank Face

Figure 6 presents the measured flank wear land for different vibration amplitudes and cutting speeds. Compared to CD, LF-VAD resulted in a lower flank wear land for all machining conditions. For the low cutting speed *N* = 2000 rpm, the tool life was enhanced by increasing $A_m$. Compared to CD at drilled hole number 50 (flank wear = 135 μm), the flank wear land was reduced by 37%, 45%, and 53% for $A_m$ = 0.1 mm, $A_m$ = 0.16 mm, and $A_m$ = 0.25 mm, respectively, as shown in Figure 6a. The reason for these reductions is the lower cutting temperature at the higher vibration amplitude, as described in Section 4.1. Compared to the dry machining condition, the MQL did not show a considerable effect on the flank wear land for CD and LF-VAD at $A_m$ = 0.25 mm. For both machining conditions, the flank wear land was reduced by 8 μm at drilled hole number 50. This ineffective

behavior of MQL at these machining conditions was assigned to the continuous tool–workpiece contact at CD, and excessive coolant time for $A_m$ = 0.25 mm. On the other hand, by up to a 50% reduction in the flank wear land, the MQL is more efficient at $A_m$ = 0.16 mm. This effect could be discussed by the relatively low cooling interval, and consequently the temperature gradient at MQL is highly recommended. Moreover, the smaller air gap width increases the MQL efficiency, as discussed in [33].

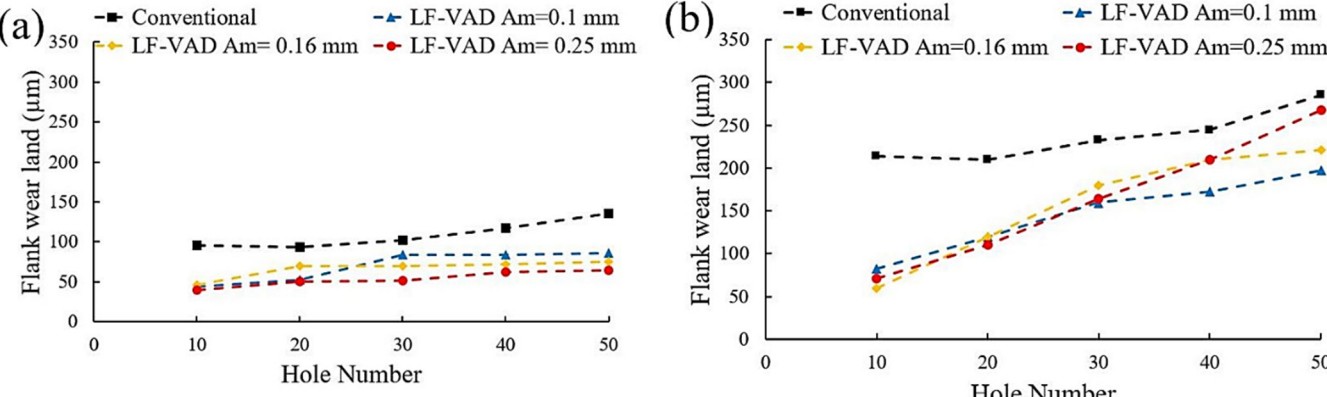

**Figure 6.** Effect of vibration amplitude on the tool flank wear land for different cutting speeds (**a**) $N$ = 2000 rpm (**b**) $N$ = 3000 rpm.

For the high cutting speed $N$ = 3000 rpm, increasing the vibration amplitude does not affect the flank wear land for the first 30 drilled hole, and a reduction of 30% was measured for all $A_m$. However, by increasing the number of drilled holes, the LF-VAD with the lowest vibration amplitude becomes favorable, as shown in Figure 6b. This effect could be the result of the higher transformation rate from the steady to severe wear regions for LF-VAD as an effect of high $A_m$ due to the relatively high thrust force, as reported in [9]. Unlike the low cutting speed, MQL showed a significant enhancement at $N$ = 3000 rpm. The MQL resulted in a successful drilling process of 50 holes with acceptable flank wear land, according to ISO 3685 ($\leq$300 μm) [34].

Furthermore, the relatively low cutting temperature and the intermitted cutting process of LF-VAD showed a significant limitation on chemical wear compared to the CD [19]. Based on the tool flank surface examination, the colored area and built-up edge were prominent in CD, while the LF-VAD showed a discolored surface with a limited built-up edge, as shown in Figure 7.

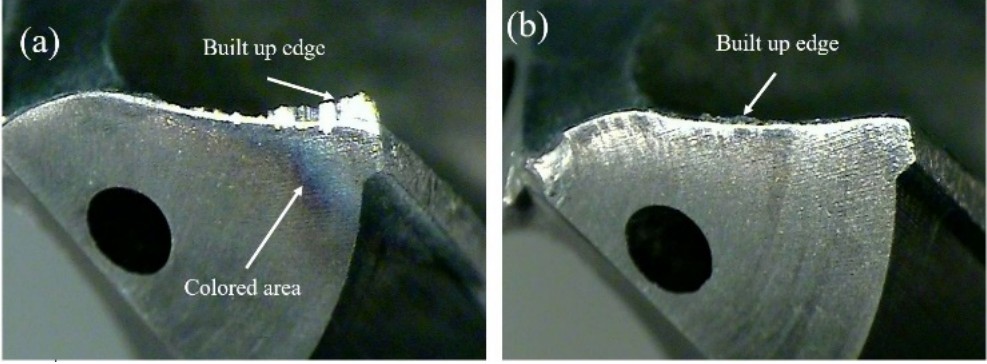

**Figure 7.** Effect of LF-VAD on the chemical wear and built-up edge formation for (**a**) conventional drill (**b**) $A_m$ = 0.16 mm.

4.2.2. Chisel Edge

The scanning electron microscopy (SEM) examination of the tool chisel edge revealed such a clear difference between the CD and LF-VAD process, as shown in Figures 8 and 9. For CD, there is obvious evidence of Ti adhesion to the chisel edge for both cutting speeds (Figures 8a and 9a). This observation was due to the relatively higher cutting temperature in CD compared to LF-VAD, as discussed in Section 4.1 due to the poor chip evacuation mechanism [21]. On the other hand, the chisel edge fracture was identified as a dominant mechanism for LF-VAD, as shown in Figure 8b,c and Figure 9b,c. As described in Section 3, the vibration amplitude was identified as a dominant effective parameter on the axial tool velocity. Increasing the vibration amplitude resulted in increasing the axial tool velocity, as shown in Figure 3. This increasing leads to a severe tool–workpiece repetitive impact mechanism. Consequently, the utilization of VAD at a high vibration amplitude could result in tool chisel edge fracture as observed. Contrary to [20], the chisel edge suffers a severe fracture rather than Ti adhesion due to the negative clearance angle. This difference could be attributed to the utilization of a higher $A_m$ range (0.1 to 0.25 mm) compared to only 0.06 mm.

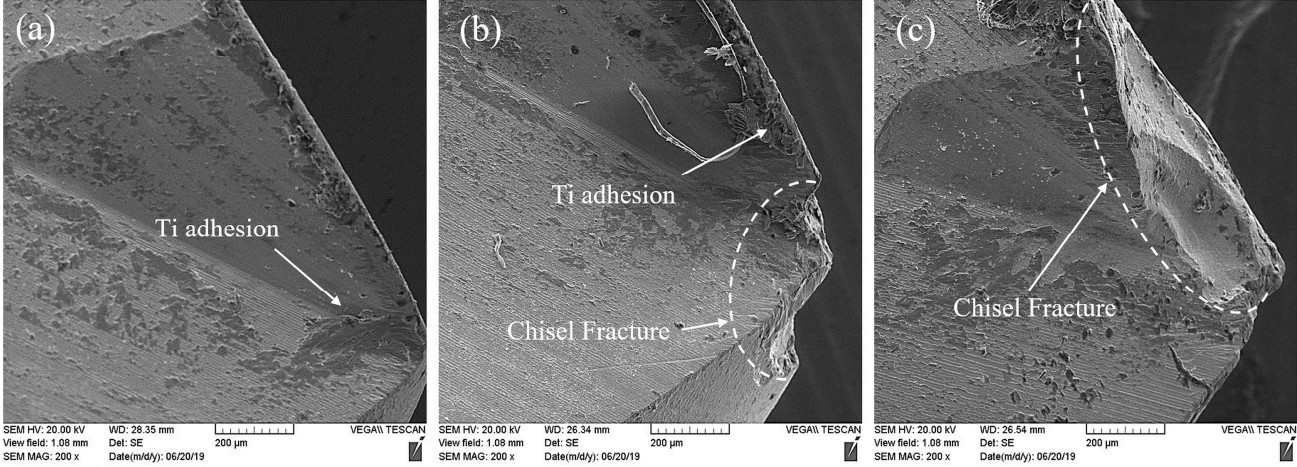

**Figure 8.** Effect of vibration amplitude on the tool chisel edge at $N$ = 2000 rpm for (**a**) conventional, (**b**) $A_m$ = 0.1 mm, (**c**) $A_m$ = 0.25 mm.

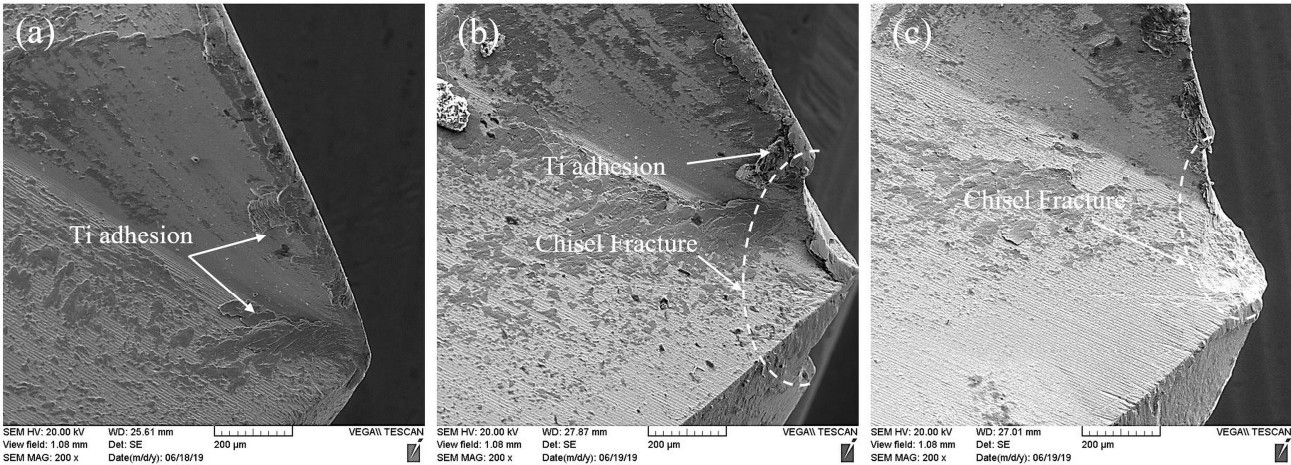

**Figure 9.** Effect of vibration amplitude on the tool chisel edge at $N$ = 3000 rpm for (**a**) conventional, (**b**) $A_m$ = 0.1 mm, (**c**) $A_m$ = 0.25 mm.

The thermal impact was identified as one of the critical drawbacks during the intermitted machining process [35,36]. Inducing the cutting tool material to a frequent temperature variation resulted in a thermal crack formation [36–38]. Consequently, increasing the temperature gradient through the utilization of a coolant medium during the LF-VAD process could increase the probability of thermal crack initiation or tool fracture. Compared to the dry machining condition [10], the MQL resulted in the observation of chisel edge fracture for both cutting speeds. This observation could be attributed to the relatively high thermal gradient during the uses of MQL. Based on the flank surface and chisel edge analysis, the dry coolant condition is highly recommended for the low cutting speed condition, while MQL is required for the higher cutting speed.

### 4.2.3. Rake Face

The surface topography was examined to identify the main tool wear mechanism for each machining condition, as shown in Figures 10 and 12. Figure 10 shows the examined rake face surface for different $A_m$ after the drilling process of an identical number of 50 holes at $N = 2000$ rpm. Adhesion and abrasion wear mechanisms were observed for all machining conditions. However, the adhesion wear mechanism had the dominant effect, especially for the CD condition. This effect was attributed to the relatively higher cutting temperature for the CD. The high chemical affinity toward most of the industrially used drill bits material highlights adhesion as a dominant wear mechanism during the drilling process of Ti materials [4,10,20,39,40]. This process resulted in a built-up edge (BUE) formation, which leads to tool fracture or chipping and consequently, poor hole quality and short tool life will be observed. Based on the surface topography analysis, the greatest height of defects was 400 μm for CD, 90 μm for $A_m = 0.1$ mm, 75 μm for $A_m = 0.16$ mm, and 160 μm for $A_m = 0.25$ mm. The LF-VAD resulted in a significant reduction in the adhered particles' height by up to 80% at $A_m = 0.16$ mm. However, increasing the vibration amplitude to $A_m = 0.25$ mm showed a relatively lower reduction percentage of 60%. This effect could be due to the higher effect of MQL turbulence or the probability of collision between the small radian chips at $A_m = 0.25$ mm [9,30], and the borehole wall. In addition, by SEM examination of the drill bit rake face used for CD, a considerable amount of carbon fiber was found adhered to the BUE, as shown in Figure 11, and confirmed by the EDS analysis in Table 2. This adhesion could be due to the tool–borehole wall interaction during the tool retraction that could have a negative effect on the CFRP geometric accuracy and surface roughness.

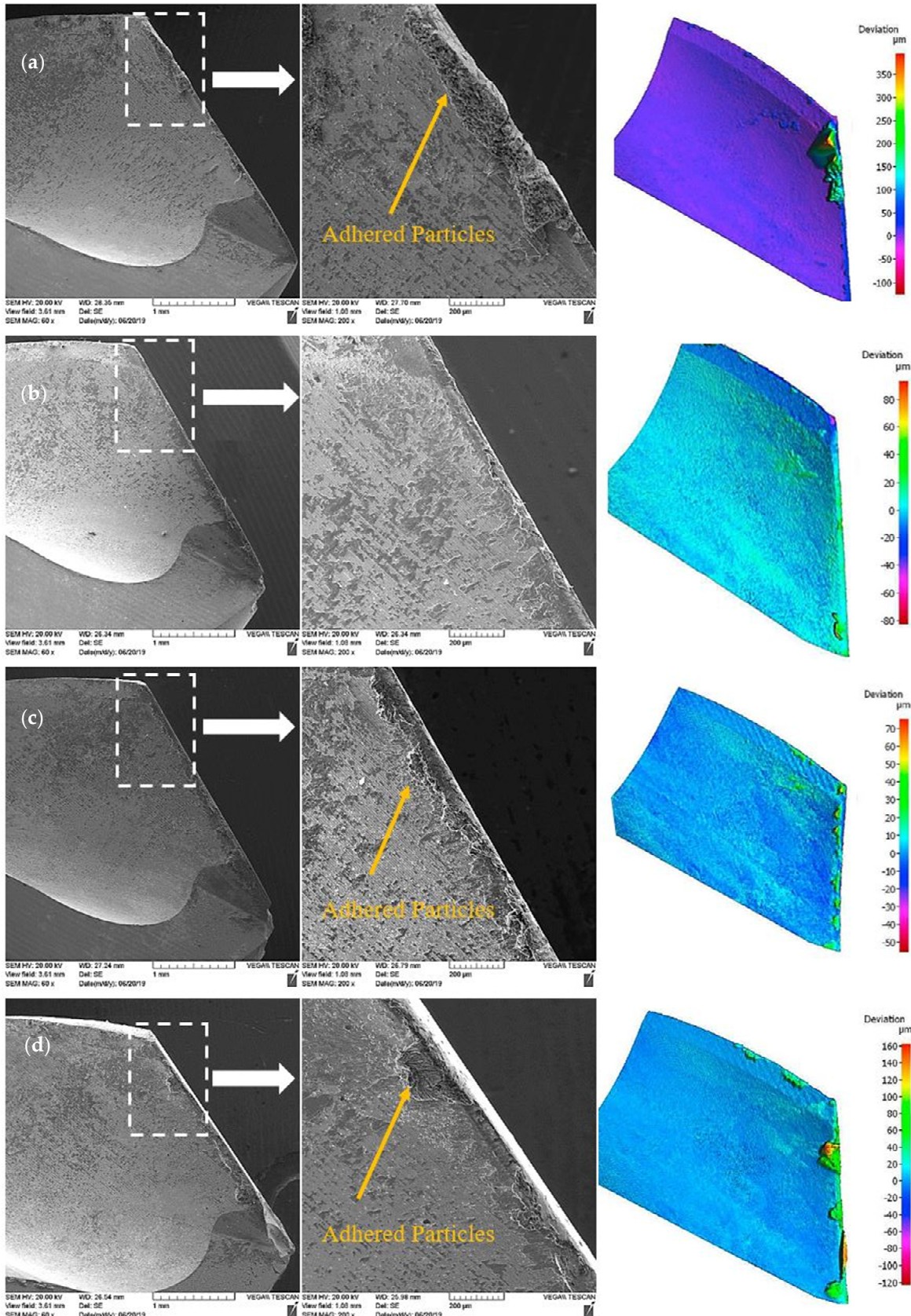

**Figure 10.** The SEM examination and surface topography analysis of the tool rake face at $N$ = 2000 rpm for different vibration amplitude (**a**) conventional, (**b**) $A_m$ = 0.1 mm, (**c**) $A_m$ = 0.16 mm, and (**d**) $A_m$ = 0.25 mm.

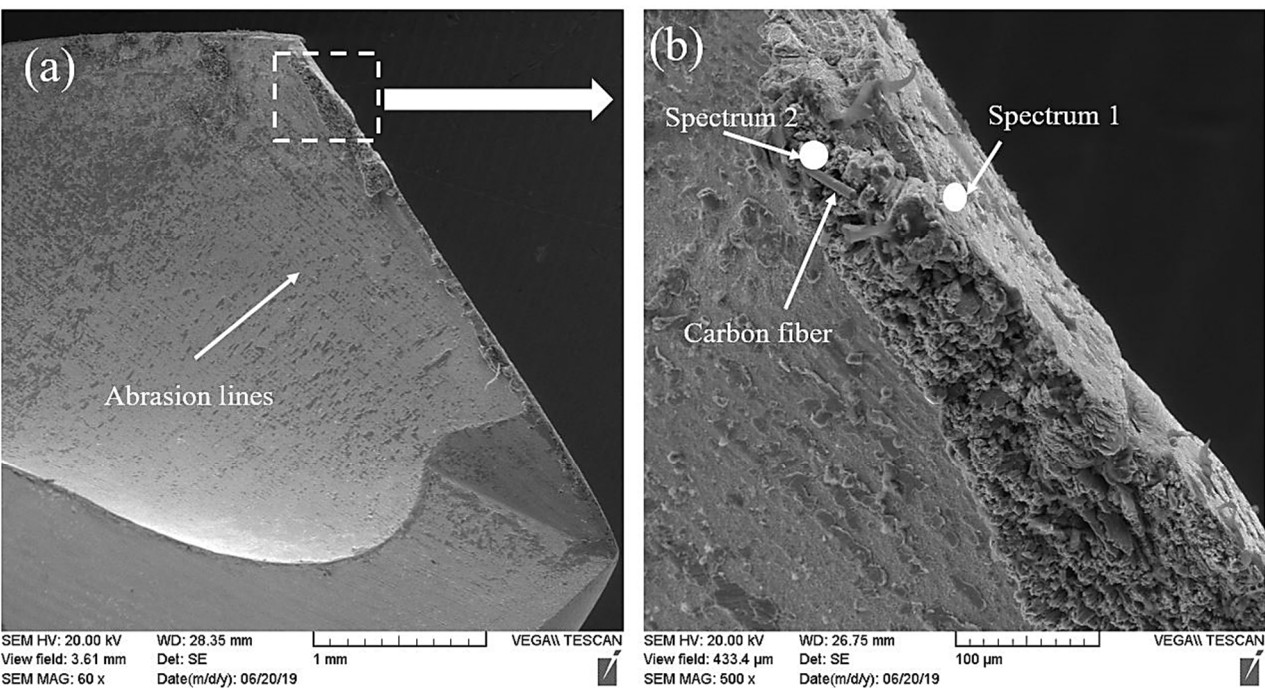

**Figure 11.** The conventional drill effect on the rake face after drilled hole number 50.

**Table 2.** The EDS analysis for the selected points, as shown in Figure 11b.

|  | C | Al | Ti | V | Co | W | Total |
|---|---|---|---|---|---|---|---|
| Spectrum 1 | 43.42 | 3.86 | 49.57 | 3.15 | 0 | 0 | 100.00 |
| Spectrum 2 | 97.43 | 0.65 | 0.47 | 0.40 | 0.25 | 0.80 | 100.00 |

Figure 12 shows the effect of different vibration amplitudes at $N = 3000$ rpm on the rake face after the drilling process of 50 identical holes. The crater wear mechanism was identified as the dominant wear for all machining conditions. This change in the tool wear mechanism was the result of the relatively higher cutting temperature at $N = 3000$ rpm, as presented in Section 4.1. The maximum crater depth was measured and plotted, as shown Figure 13. Compared to the CD, the LF-VAD resulted in a lower crater depth with up to 33% reduction at $A_m = 0.25$ mm, as shown in Figure 13. The lower crater depth for LF-VAD at high $A_m$ was attributed to the lower cutting temperature, as discussed in Section 4.1. Moreover, increasing the $A_m$ resulted in a significant reduction in the tool–chip contact time, as presented in [9]. This reduction had a positive impact on restricting the adhesion wear on the tool rake surface.

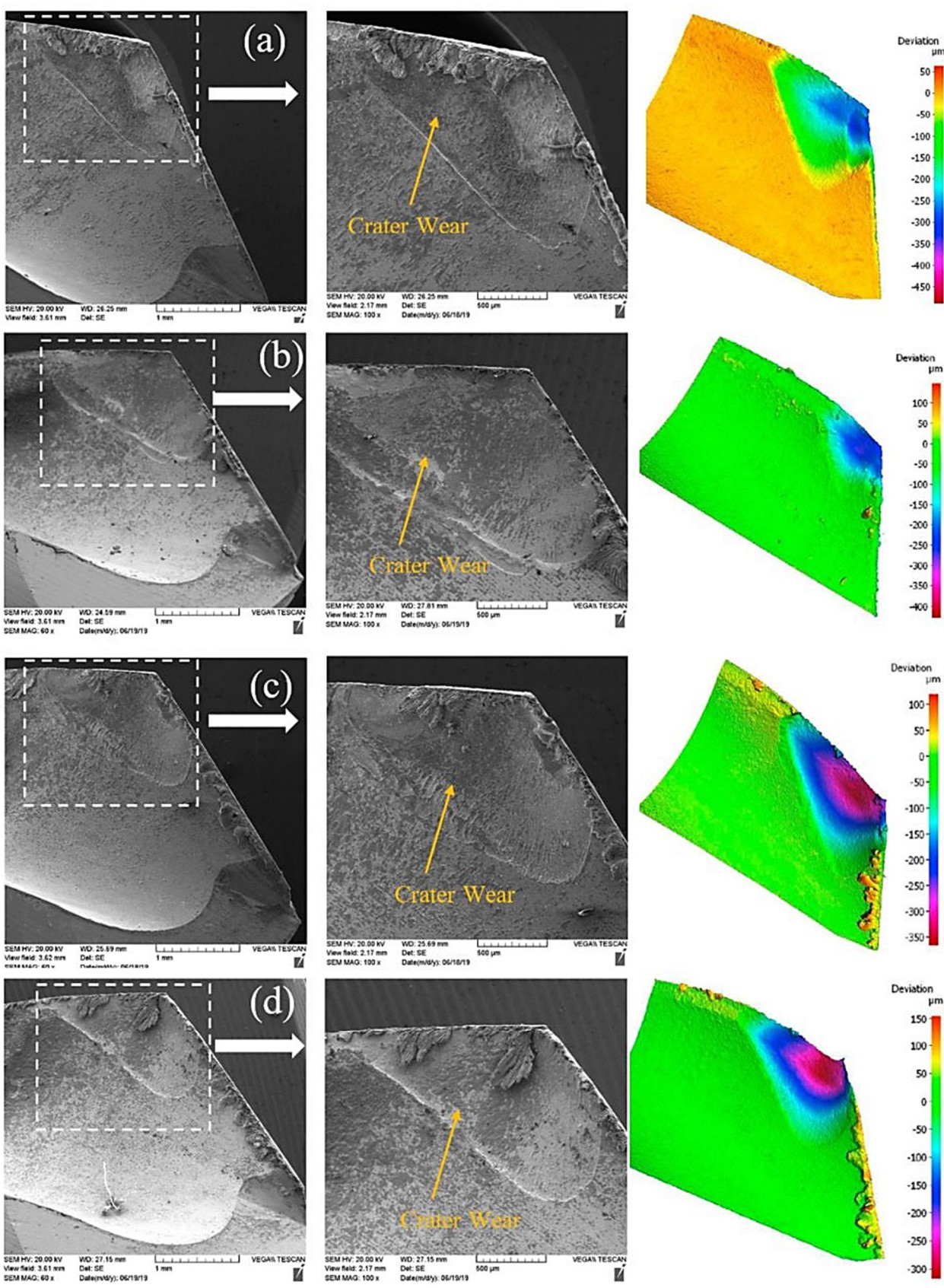

**Figure 12.** The SEM examination and surface topography analysis of the tool rake face at $N$ = 3000 rpm for different vibration amplitude (**a**) conventional, (**b**) $A_m$ = 0.1 mm, (**c**) $A_m$ = 0.16 mm, and (**d**) $A_m$ = 0.25 mm.

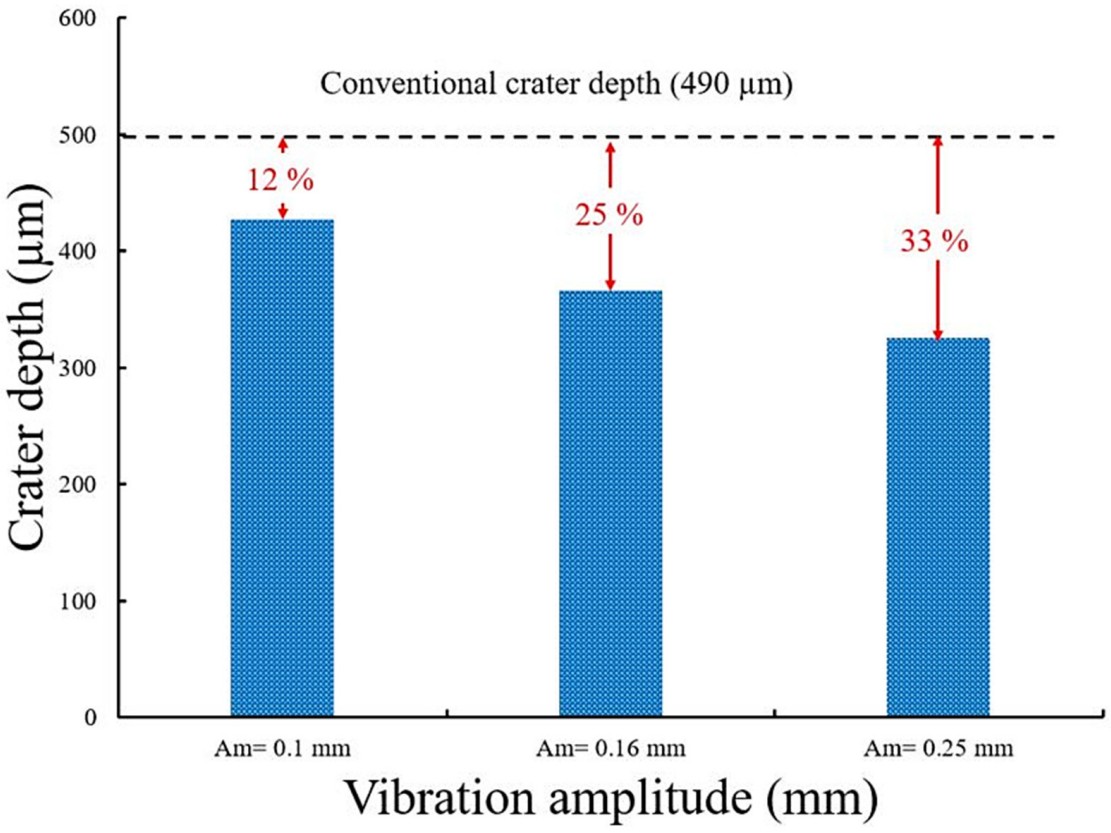

**Figure 13.** The measured crater depth at different vibration amplitude for *N* = 3000 rpm.

### 4.3. Effect of Tool Wear on the Exit Delamination

Based on the CFRP exit surface examination, the drilling process of CFRP/Ti6Al4V commonly results in a discoloration ring and a damaged area [41]. The discoloration ring was caused due to the cutting temperature at the CFRP/Ti6Al4V interface surface, while the Ti6Al4V chips evacuation mechanism resulted in severe damage of the borehole wall. Table 3 presents the effect of tool wear on the exit delamination for different vibration amplitudes and cutting speeds. The delamination factor ($\varnothing_d$) was identified based on the following equation [19,42]:

$$\varnothing_d = \frac{D_{actual} - D_{nominal}}{D_{nominal}} \tag{4}$$

where *D* actual is the diameter of a circle including the discoloration ring and all damaged area, while *D* nominal represents the nominal hole diameter.

**Table 3.** The effect of tool wear on the exit delamination for different vibration amplitudes and cutting speeds.

| Cutting Speed (rpm) | Vibration Amplitude (mm) | Drilled Hole Number | | | | |
|---|---|---|---|---|---|---|
| | | 0–10 | 10–20 | 20–30 | 30–40 | 40–50 |
| N = 2000 | Conventional | | | 0.1–0.2 | | |
| | $A_m$ = 0.1 | | | Free | | |
| | $A_m$ = 0.16 | | | | | |
| | $A_m$ = 0.25 | Free | | 0.05–0.1 | | |
| N = 3000 | Conventional | 0.05–0.1 | | 0.1–0.2 | | 0.2–0.3 |
| | $A_m$ = 0.1 | | | Free | | |
| | $A_m$ = 0.16 | | | | | |
| | $A_m$ = 0.25 | | | 0.05–0.1 | | |

For all machining conditions, the measured $\varnothing_d$ was acceptable, as defined by aerospace manufacturers ($\varnothing_d \leq 0.5$) [18]. However, free exit delamination was successfully achieved

by using LF-VAD with $A_m$ = 0.1 mm and 0.16 mm, as shown in Figure 14. This observation was attributed to the lower cutting temperature and the proper chip evacuation efficiency. The slight increase in $\varnothing_d$ for $A_m$ = 0.25 mm could be due to the negative effect of dynamic tool movement at this amplitude. As described in Section 3, the axial tool velocity reached the maximum value at a 0.25 mm vibration amplitude. This high axial velocity could increase the friction force at the tool–hole wall interface. Hence, more thermal defects could be observed and alter the delamination factor.

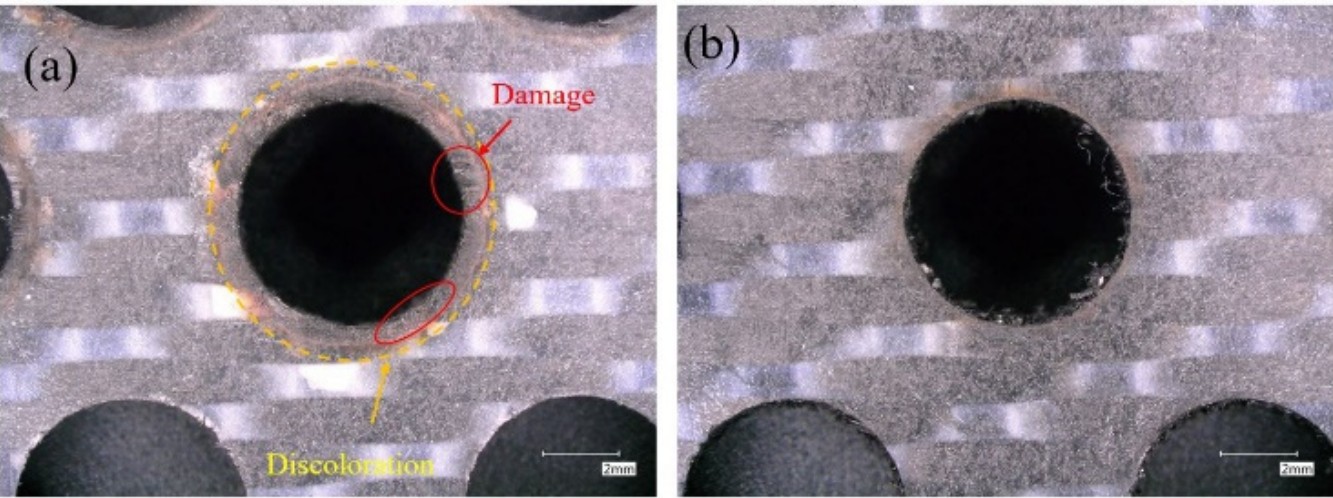

**Figure 14.** The effect of LF-VAD on the exit delamination for *N* = 3000 rpm (**a**) conventional (**b**) $A_m$ = 0.16 mm.

### 4.4. Effect of Tool Wear on the Geometrical Accuracy

Based on the aerospace manufacturers' recommendations, the acceptable hole size error was identified from −0.7% to 0.4% [18]. For both cutting speeds, CD exceeds the acceptable CFRP hole size, as shown in Figure 15a,b. This unacceptable hole size was a result of the destructive effect of Ti6Al4V chips during evacuation. On the contrary, the LF-VAD with $A_m$ = 0.16 mm and 0.25 mm resulted in an acceptable accuracy for *N* = 2000 rpm and the first 25 drilled hole at *N* = 3000 rpm. The unacceptable hole accuracy after drilled hole number 25 could be reverted to the negative effect of flank wear land progression. In addition, the LF-VAD with $A_m$ = 0.1 mm resulted in a relatively higher chip radian [9] that could reduce the chips evacuation efficiency, thus increasing the negative damage effect on the CFRP wall. On the other hand, both machining processes resulted in an acceptable Ti6Al4V hole accuracy for all drilled holes at different cutting speeds, as shown in Figure 15c,d. However, the LF-VAD with $A_m$ = 0.16 mm and 0.25 mm showed the lowest geometric deviation for both cutting speeds.

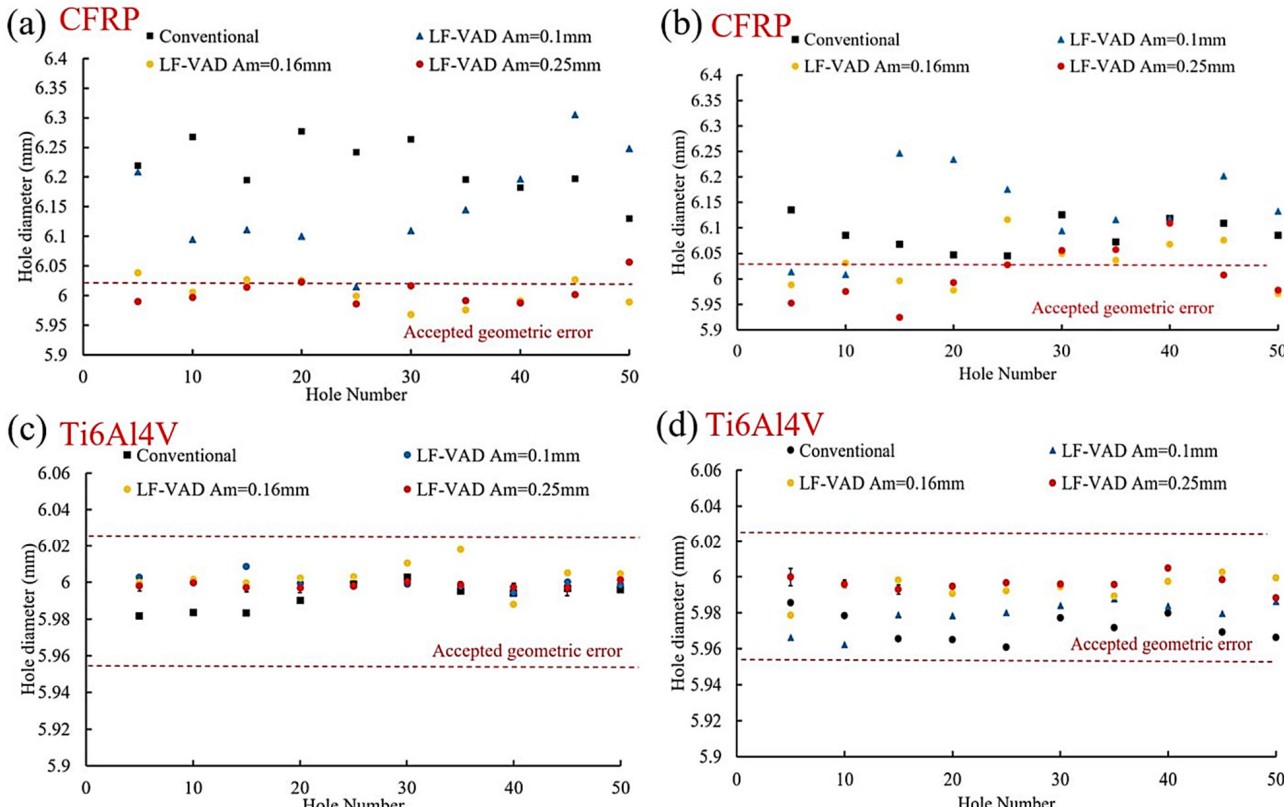

**Figure 15.** Effect of vibration amplitude on the CFRP and Ti6Al4V hole accuracy at different cutting speeds (**a**,**c**) N = 2000 rpm, (**b**,**d**) N = 3000 rpm.

### 4.5. Burr Height

Thermal load and thrust force are of the main machining characteristics that control the material burr formation at the exit surface during the drilling process. For CD, the burr height increased from 0.4 mm at the first drilled hole, to 0.7 mm at drilled hole number 50, as shown in Figure 16a,b. This increase was attributed to the higher cutting temperature, as discussed in Section 4.1. For LF-VAD at the first drilled hole, the burr height was 1.2 mm and 1.36 mm for $A_m$ = 0.1 mm and $A_m$ = 0.25 mm, respectively. Increasing the drilled hole number with LF-VAD showed a positive effect, compared to CD. The measured burr height at drilled hole number 50 was reduced to 1.04 mm and 0.57 mm for $A_m$ = 0.1 mm and $A_m$ = 0.25 mm, respectively as shown in Figure 16c,d. This reduction could be attributed to the chisel edge fracture as observed and discussed in Section 4.2.2. This fracture increases the tool–workpiece contact area under the chisel edge that resulted in reducing the applied stresses magnitude on the machined material, and consequently a lower driving force for burr formation. Furthermore, maintaining a low cutting temperature for $A_m$ = 0.25 mm resulted in a further burr height reduction compared to $A_m$ = 0.1 mm, as shown in Figure 16d,f.

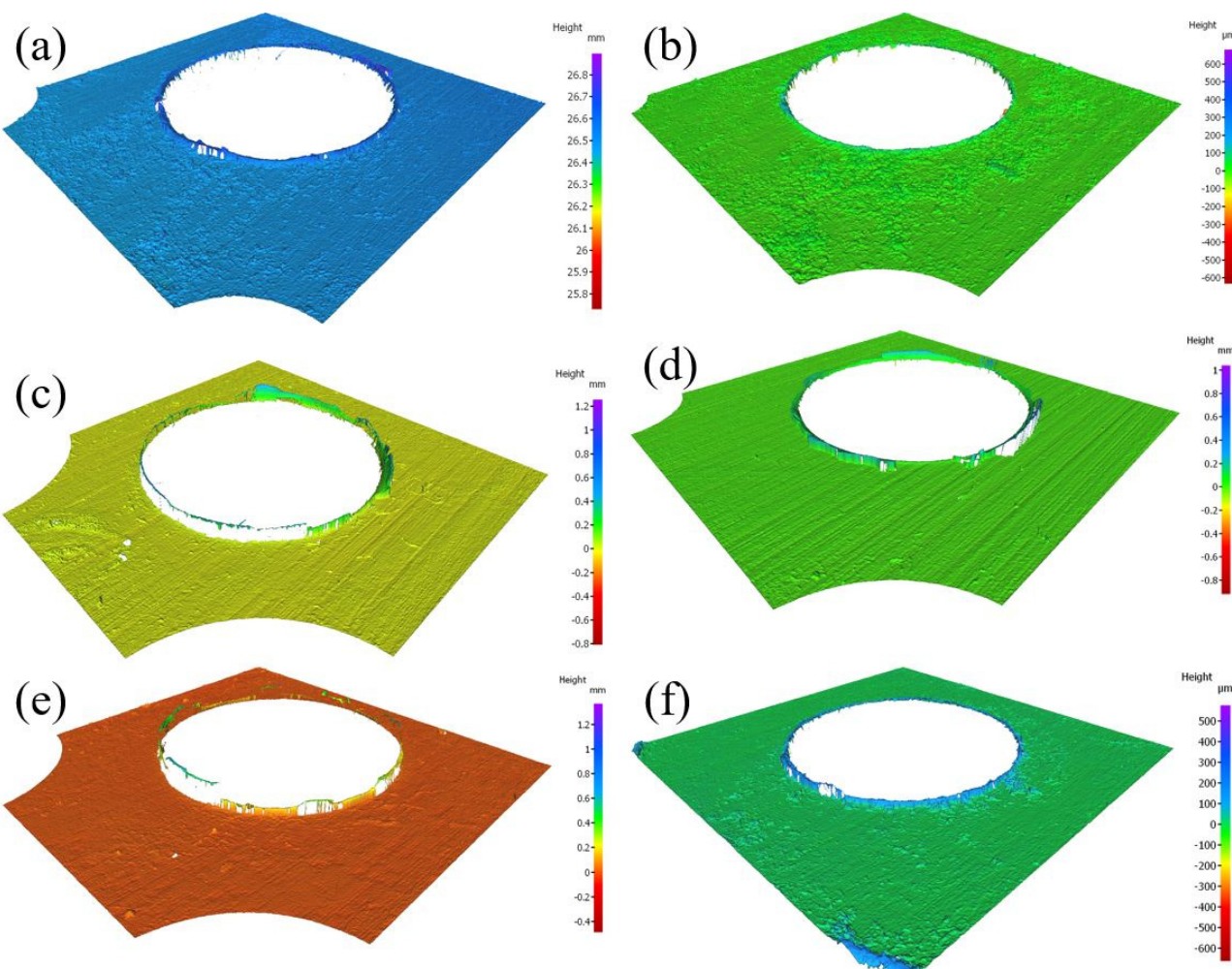

**Figure 16.** Effect of tool wear progression on the burr height at $N$ = 3000 rpm for different machining parameters: (**a**) CD at first hole, (**b**) CD at hole number 50, (**c**) $A_m$ = 0.1 mm at first hole, (**d**) $A_m$ = 0.1 mm at hole number 50, (**e**) $A_m$ = 0.25 mm at first hole, and (**f**) $A_m$ = 0.25 mm at hole number 50.

### 4.6. MQL vs. Dry Coolant Condition

Based on the exit delamination analysis during the dry machining condition [10] and the MQL coolant medium, Table 4 presents the effect of each coolant condition on the observed machining process at $N$ = 2000 rpm. For the CD, MQL showed a significant enhancement on the exit delamination factor and Ti hole accuracy. However, the geometrical accuracy of CFRP has an unacceptable range for both machining conditions. On the other hand, MQL is only useful for LF-VAD at $N$ = 3000 rpm, while at low $N$ = 2000 rpm, both machining conditions have the same exit delamination factor.

**Table 4.** Effect of coolant condition on the CFRP/Ti6Al4V drilling process at $N$ = 2000 rpm.

| Coolant Condition | Dry | | | | MQL | | | |
|---|---|---|---|---|---|---|---|---|
| Amplitude ($A_m$) | **CD** | **0.1 mm** | **0.16 mm** | **0.25 mm** | **CD** | **0.1 mm** | **0.16 mm** | **0.25 mm** |
| Flank wear land | ≥300 µm | ≤300 µm | | | ≤300 µm | | | |
| Chisel edge | No fracture | | | | No fracture | Fracture | | |
| Exit delamination | Hole No ≤20 | Acceptable exit delamination factor | | | Acceptable exit delamination factor | | | |
| CFRP diameter accuracy | No | N/A | | Yes | No | No | | Yes |
| Ti diameter accuracy | No | N/A | | Yes | Yes | Yes | | |

## 5. Conclusions

This paper addressed the previous challenges facing the low frequency vibration-assisted drilling (LF-VAD) of aerospace CFRP/Ti6Al4V stacked material through the utilization of minimum quantity lubricant (MQL) cooling condition. The experimental investigation concerns the effect of LF-VAD coupled with MQL on the tool wear progression, cutting temperature, and hole geometry. The following points highlight the main conclusions:

- Compared to the LF-VAD under the dry condition, MQL increased the applicable feed rate from 0.025 mm/rev to 0.075 mm/rev at $N$ = 3000 rpm. The coupled effect of LF-VAD and MQL showed a higher accuracy hole geometry and longer tool life owing to a proper chip evacuation mechanism and optimum cooling condition. This increased feed rate improved machining productivity by 300%.
- Vibration amplitude and cooling medium are critical to the cutting temperature, which showed a significant reduction by up to 65%, compared to the CD dry drilling of CFRP/Ti6Al4V stacks. This reduction contributed to the smaller chip radian, higher tool axial velocity, lower fractional force at the tool–hole wall interface, and the advance of smaller micro coolant droplets to penetrate the tool–chip and tool–workpiece interface. The MQL assistance resulted in a drilling process of 50 holes without any observation of the tool–chip welding phenomenon.
- For all experimental investigations, LF-VAD showed a flank wear land reduction by up to 53%, due to the lower cutting temperature. The flank wear land has a critical impact on tool life evaluation and the machining productivity.
- From the tool examination, the chisel edge was identified as the weakest point for LF-VAD. This observation can be traced back to the repetitive tool–workpiece impact mechanism. On the other hand, the tool outer corner was identified as the most vulnerable area for catastrophic failure during CD.
- The LF-VAD resulted in a reduction of 80% of the maximum BUE height at $N$ = 2000 rpm, while the crater depth was reduced by 33% at $N$ = 3000 rpm.
- Under the MQL condition, LF-VAD produces more consistent hole diameters and no CFRP exit delamination.

**Author Contributions:** R.H. performed experiments, analysis, and data interpretation; wrote the first draft of the manuscript; helped with submitting the final manuscript to the journal (corresponding author). A.S. helped with the experiments and analysis; revised the manuscript. M.A.E. revised and edited the manuscript and gave the final approval to be submitted. H.A. revised the manuscript. All authors have read and agreed to the published version of the manuscript.

**Funding:** This research received no external funding.

**Institutional Review Board Statement:** Not applicable.

**Informed Consent Statement:** Not applicable.

**Data Availability Statement:** The datasets generated during and/or analysed during the current study are currently not publicly available due to their use in an ongoing research but are available from the corresponding author on reasonable request.

**Conflicts of Interest:** The authors declare no conflict of interest.

## Abbreviations

| | |
|---|---|
| VAD | Vibration-assisted drilling |
| LF-VAD | Low frequency vibration-assisted drilling |
| CD | Conventional drilling |
| BUE | Built-up edge |
| MQL | Minimum quantity lubricant |

## Notations

| | |
|---|---|
| $N$ | Cutting speed (rpm) |
| $f$ | Feed rate (mm/rev) |
| $A_m$ | Modulation amplitude (mm) |
| $F$ | Frequency (oscillation/rev) |

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
