# Peer review of "The Effect of MQL on Tool Wear Progression in Low-Frequency Vibration-Assisted Drilling of CFRP/Ti6Al4V Stack Material"

_jmmp, doi:10.3390/jmmp5020050_

Round 1

Reviewer 1 Report

This work is interesting regarding the study of tool wear progression in vibration-assisted drilling of carbon fiber reinforced polymer. The introduction and methodology are clearly presented; however, the kinematics of the VAD is not clearly described, the quality of several figures is poor and there is no novelty on how experimental results could be used to predict variables of interest.

Major Comments:

  • All references for figures are broken, which makes it difficult to understand. The authors should not allow themselves this kind of error during the submission process.
  • The effect of vibration amplitude on the kinematics of the VAD is not well described. The authors do not present any correlation between the VAD parameters and experimental results. There is enough experimental data that could be used, for example, to propose a mechanistic model for wear, cutting forces, or other parameters under different cutting conditions.
  • The figure quality is not good enough for most of them (e.g. Figure 4). The font size is too small to be legible. Take into consideration that font size should be similar text.
  • Conclusions should be improved.

Minor comments

  • Figure 17 need to be converted to a table.
  • Remove all information from the bottom in Figure 1. Is there any relevant information that needs to be included? In that case, increase the font size.
  • It is not possible to identify the numbers for the color bars in Figures 3, 5, 9.

Author Response

The authors would like to thank the reviewers for the valuable comments and careful review of the manuscript entitled “The effect of MQL on tool wear progression in low-frequency vibration-assisted drilling of CFRP/Ti6Al4V stack material”. The authors appreciate the time taken to review the article.

Reviewer #1:

Q1

·        All references for figures are broken, which makes it difficult to understand. The authors should not allow themselves this kind of error during the submission process.

A1

The authors agree with the reviewer comment. The following changes added to the manuscript:

·       All the figure references have been checked.

·       All the figures and tables numbering have been checked in the manuscript.

Q2

·        The effect of vibration amplitude on the kinematics of the VAD is not well described. The authors do not present any correlation between the VAD parameters and experimental results. There is enough experimental data that could be used, for example, to propose a mechanistic model for wear, cutting forces, or other parameters under different cutting conditions.

A2

The kinematics of VAD section were improved, and the kinematics effect on the cutting temperature, tool wear (chisel edge), and delamination analysis have been added to the manuscript.  

Q3

·        The figure quality is not good enough for most of them (e.g. Figure 4). The font size is too small to be legible. Take into consideration that font size should be similar text.

A3

·        All the figures will resubmitted to the journal using anther format to enhance the quality and Figure 4 font size has been checked.

Q4

·        Conclusions should be improved.

A4

·        The authors agree with the reviewer comment, and the conclusion has been improved.

Q5

·        Figure 17 need to be converted to a table.

A5

·        Figure 17 has been converted to table 4 in the manuscript.

Q6

·        Remove all information from the bottom in Figure 1. Is there any relevant information that needs to be included? In that case, increase the font size.

A6

·        The Figure was changed and represented without information.

Q7

·        It is not possible to identify the numbers for the color bars in Figures 3, 5, 9.

·         

A7

·        Figures 3,5,6 have been adjusted and added to the manuscript.

Reviewer 2 Report

The reseach work is well organised and the experimental and evaluation procedure is adequate for extracting realisting results. The question which is not answered, is how these results can really improve the cutting process and how manufacturer can benefit from them in order to increase their prodactuvity.  

Author Response

The authors would like to thank the reviewers for the valuable comments and careful review of the manuscript entitled “The effect of MQL on tool wear progression in low-frequency vibration-assisted drilling of CFRP/Ti6Al4V stack material”. The authors appreciate the time taken to review the article.

Reviewer #2:

Q1. The question which is not answered, is how these results can really improve the cutting process and how manufacturer can benefit from them in order to increase their productivity.  

A1. The conclusion has been improved to answer the reviewer question and present impact of this paper can improve machining performance and productivity.

Reviewer 3 Report

The reviewer comments of the paper «The effect of MQL on tool wear progression in low-frequency vibration-assisted drilling of CFRP/Ti6Al4V stack material»- Reviewer

The authors presented an article «The effect of MQL on tool wear progression in low-frequency vibration-assisted drilling of CFRP/Ti6Al4V stack material». Reviewed article is very interesting and write at good scientific level. However, there are several points in the article that require further explanation.

Comment 1:

The introduction needs to be improved and a more detailed analysis of the reference should be provided.

First, group citations should be excluded. Maximum one, two ctitations in one phrase [..., ...]. Try to split the sentence into several or use qualifying phrases. ... [...], ... [...] etc. A more detailed analysis of each reference should be made. 1 reference = 1 sentence.

However, it is helpful to add an article: International Journal of Advanced Manufacturing Technology 2018, 98(10), 2801–2817. doi:10.1007/s00170-018-2410-2

Secondly, a more detailed analysis of cooling methods should be done. Consider the article:

Materials, 2021, 14(4), 795. doi:10.3390/ma14040795

Wear 2019, 426-427, 1616-1623. doi:10.1016/j.wear.2019.01.005

Journal of Materials Research and Technology 2021, 11, 719–753. doi:10.1016/j.jmrt.2021.01.031.

Comment 2:

  1. Experimental setup

For devices, software and machines used in research, indicate in parentheses (manufacturer, city, country).

Give more information on the structure and composition of the material (CFRP and Ti6Al4V). It is helpful to give SEM analysis of the microstructure.

Are all figures original? If not needed appropriate citations and permissions.

Are all formulas original? If not needed appropriate citations.

Comment 3:

  1. Results and discussion

The resolution and quality of the figures needs to be improved.

Moreover, authors should check the numbering of all figures and tables. There are many inconsistencies. Tables 1 and 3 are shown as fig. Table 2 not found?

Comment 4:

Conclusions

In addition, it is necessary to more clearly show the novelty of the article and the advantages of the proposed method. What is the difference from previous work in this area? Show practical relevance.

The article is interesting. However, the article needs to be improved. Authors should carefully study the comments and make improvements to the article step by step. All changes should be highlighted in color. After major changes can an article be considered for publication in the «Journal of Manufacturing and Materials Processing».

Author Response

The authors would like to thank the reviewers for the valuable comments and careful review of the manuscript entitled “The effect of MQL on tool wear progression in low-frequency vibration-assisted drilling of CFRP/Ti6Al4V stack material”. The authors appreciate the time taken to review the article.

Reviewer #3:

The authors want to thank the reviewer so much for his comments and appreciate his comments.

Q1

The introduction needs to be improved and a more detailed analysis of the reference should be provided.

First, group citations should be excluded. Maximum one, two ctitations in one phrase [..., ...]. Try to split the sentence into several or use qualifying phrases. ... [...], ... [...] etc. A more detailed analysis of each reference should be made. 1 reference = 1 sentence.

However, it is helpful to add an article: International Journal of Advanced Manufacturing Technology 2018, 98(10), 2801–2817. doi:10.1007/s00170-018-2410-2

Secondly, a more detailed analysis of cooling methods should be done. Consider the article:

Materials, 2021, 14(4), 795. doi:10.3390/ma14040795

Wear 2019, 426-427, 1616-1623. doi:10.1016/j.wear.2019.01.005

Journal of Materials Research and Technology 2021, 11, 719–753. doi:10.1016/j.jmrt.2021.01.031.

     A1

The authors agree with the reviewer comment. The following changes added to the manuscript:

·       All the group citations have been excluded.

·       More detailed analysis of cooling methods have been added to the manuscript  Line 90-109 and Line 115-119

Q2

For devices, software and machines used in research, indicate in parentheses (manufacturer, city, country).

Give more information on the structure and composition of the material (CFRP and Ti6Al4V). It is helpful to give SEM analysis of the microstructure.

Are all figures original? If not needed appropriate citations and permissions.

Are all formulas original? If not needed appropriate citations.

A2

1-    The manufacturer information for all devices, software and machines have been added to the experimental section.

2-    The Ti6Al4V grade has been added, while the CFRP structure information is fully described in table 1.

3-    All the figures are original.

4-     Regarding the formulas, appropriate citation was added.

Q3

The resolution and quality of the figures needs to be improved.

Moreover, authors should check the numbering of all figures and tables. There are many inconsistencies. Tables 1 and 3 are shown as fig. Table 2 not found?

A3

The authors agree with the reviewer comment. The following changes added to the manuscript:

·       All the figures will resubmitted to the journal using anther format.

·       All the figures and tables numbering have been checked in the manuscript.

·       All the tables have been checked and adjusted in the manuscript.

Round 2

Reviewer 1 Report

The authors have addressed all my concerns. From my side, the paper can be published in MDPI JMMP.

Reviewer 3 Report

The authors have improved the article according to the comments. The article can now be published.